# Sirtuin 1 and Skin: Implications in Intrinsic and Extrinsic Aging—A Systematic Review

**DOI:** 10.3390/cells10040813

**Published:** 2021-04-06

**Authors:** Angelika Bielach-Bazyluk, Edyta Zbroch, Hanna Mysliwiec, Alicja Rydzewska-Rosolowska, Katarzyna Kakareko, Iwona Flisiak, Tomasz Hryszko

**Affiliations:** 1Department of Dermatology and Venereology, Medical University of Bialystok, 15-540 Bialystok, Poland; hanna.mysliwiec@umb.edu.pl (H.M.); iflisiak@umb.edu.pl (I.F.); 22nd Department of Nephrology and Hypertension with Dialysis Unit, Medical University of Bialystok, 15-276 Bialystok, Poland; alicja.rosolowska@umb.edu.pl (A.R.-R.); katarzyna.kakareko@umb.edu.pl (K.K.); tomasz.hryszko@umb.edu.pl (T.H.); 3Department of Internal Medicine and Hypertension, Medical University of Bialystok, 15-540 Bialystok, Poland; edyta.zbroch@umb.edu.pl

**Keywords:** sirtuin 1, skin, aging, photoaging

## Abstract

Skin, as the outermost organ of the body, is constantly exposed to both intrinsic and extrinsic causative factors of aging. Intrinsic aging is related to compromised cellular proliferative capacity, and may be accelerated by harmful environmental influences with the greatest significance of ultraviolet radiation exposure, contributing not only to premature aging, but also to skin carcinogenesis. The overall skin cancer burden and steadily increasing global antiaging market provide an incentive for searching novel targets to improve skin resistance against external injury. Sirtuin 1, initially linked to extension of yeast and rodent lifespan, plays a key role in epigenetic modification of proteins, histones, and chromatin by which regulates the expression of genes implicated in the oxidative stress response and apoptosis. The spectrum of cellular pathways regulated by sirtuin 1 suggests its beneficial impact on skin aging. However, the data on its role in carcinogenesis remains controversial. The aim of this review was to discuss the relevance of sirtuin 1 in skin aging, in the context of intrinsic factors, related to genetic premature aging syndromes, as well as extrinsic modifiable ones, with the assessment of its future application. PubMed were searched from inception to 4 January 2021 for relevant papers with further search carried out on ClinicalTrials.gov. The systematic review included 46 eligible original articles. The evidence from numerous studies proves sirtuin 1 significance in both chronological and premature aging as well as its dual role in cancer development. Several botanical compounds hold the potential to improve skin aging symptoms.

## 1. Introduction

Aging is a continuous inherent process of gradual decline in bodily functions, and physical and mental capacity, accompanied by decreased fertility and increased susceptibility to diseases, eventually leading to death. Although the process is genetically determined, various internal and external stimuli can modify its course [1]. Thus, aging processes could be categorized as: intrinsic related to genetic, metabolic, and hormonal factors; or extrinsic, referring to external agents associated with environment and lifestyle. In most cases, extrinsic factors are modifiable, while intrinsic aging remains almost unalterable.

Skin, as the outermost organ of the body, is constantly exposed to hazardous chemical and physical agents, contributing to accelerated aging and cancer development. Aging of the world’s population and rising emphasis on maintaining physical appearance results in increased demand for antiaging products, devices, and medical procedures [2]. On the other hand, in 2018 almost 300,000 people were diagnosed with melanoma, while the incidence of non-melanoma skin cancers was over 1 million cases [3]. The global public health burden of skin tumors is even greater due to underestimated incidence rate of non-melanoma skin cancer associated with incomplete reporting. From the standpoint of individuals, global anti-aging market, and public health, it is favorable to search for novel strategies fighting against premature skin aging and carcinogenesis.

One of the promising targets are proteins belonging to sirtuin family. Sirtuin 1 (SIRT1), the most extensively studied member of the family, is NAD^+^-dependent deacetylase, which regulates numerous biological pathways by promoting chromatin silencing and transcriptional repression in dependence to energetic state of the cell [4]. Energy restriction increases the NAD^+^/NADH ratio, and subsequently the SIRT1 activity, resulting in prolonged lifespan in yeast, rodent, and human culture models [5,6,7,8]. Several organic compounds—naturally occurring abundantly in vegetables, fruits, and wine—can mimic the effect of caloric restriction (CR) on SIRT1 expression [9,10,11]. Sirtuin 1 is implicated in oxidative stress response, inflammation, mitochondrial function, and thereby in the development of several age-related diseases [12]. It regulates transcriptional activities of several proteins engaged in the oxidative stress response, increases mitochondrial number and respiratory function through peroxisome proliferator-activated receptor-γ coactivator 1α (PGC-1α) [13]. A great deal of research has shown that SIRT1 promotes cell survival by inhibiting apoptosis, and thus, for several decades it has been recognized as a tumor promoter [14]. Oppositely, the fact that further studies have displayed reduced SIRT1 level in some types of cancer together with its relevant role in counteracting oxidative stress and maintaining genome integrity, suggested that it may also act as tumor suppressor [15].

## 2. Objective

This systematic review addresses the issue of sirtuin 1 relevance in skin aging in the context of intrinsic factors related to genetic premature aging syndromes as well as extrinsic modifiable ones with the assessment of its future application. It aims to answer the question if sirtuin 1 holds the potential for prevention of premature aging or cutaneous carcinogenesis. The review includes experimental and interventional studies on animals and humans.

## 3. Materials and Methods

A systematic literature search of MEDLINE through PubMed and ClinicalTrials.gov, conducted on 4 January 2021, was performed using relevant medical subject headings (MeSH) with all subheadings included and without date limitations. The main subject of the research was sirtuin 1 relevance with regard to skin disorders, especially skin aging and neoplasms, as well as premature aging syndromes. Additionally, we reviewed the significance of environmental and lifestyle factors, considered to affect sirtuin 1 expression based on previous literature. MeSH terms included: ‘sirtuin 1′, ‘skin’, ‘skin aging’, ‘skin neoplasms’, ‘air pollution’, ‘ultraviolet ray’, ‘particulate matter’, ‘environmental pollutants’, ‘dietary supplements’, ‘diet therapy’, ‘gene activation’, ‘xeroderma pigmentosum’, ‘ataxia telangiectasia’, ‘Bloom syndrome’, ‘Werner syndrome’, ‘Cockayne syndrome’, and ‘progeria’. The terms were combined individually using the Boolean operator ‘AND’. ClinicalTrials.gov was explored with keywords ‘SIRT1′ and ‘aging’ to identify ongoing or completed trials. Non-English publications were excluded from the analysis. Due to scarcity of data experimental, animal, and human original studies were included into systematic review. The results of search strings were merged together, and duplicates were removed. Afterwards, titles and abstracts of the remaining studies were independently screened by two reviewers (A.B.-B. and E.Z.) in order to identify relevant articles that address the review subject. Disagreements between reviewers were resolved by the opinion of the third reviewer (H.M.) Finally, selected eligible articles were fully reviewed.

## 4. Results

The search resulted in the retrieval of 589 records of which 545 were screened for relevance and 46 ultimately included in qualitative synthesis. ClinicalTrials.gov identified seven interventional studies, however, after screening none of them were eligible for the review. The flowchart presenting full search strategy is demonstrated in Figure 1.

The basic characteristics of the original papers on the researched topic are summarized in Table 1, Table 2, Table 3 and Table 4. We have identified 37 experimental and 16 human observational studies of which seven were simultaneously conducted in vitro and in humans. Therefore, the assessment of risk of bias and applicability concerns were non-essential.

## 5. Discussion

Since the discovery that silent information regulator 2 (Sir2) mediates the effect of caloric restriction on lifespan extension in yeast, many attempts have been made to establish if it translates to mammals [6,62]. Extensive research on sirtuin 1, the closest structural Sir2 homologue, suggest that it may extends human lifespan indirectly by the attenuation of age-related and metabolic diseases [63,64,65,66,67]. Structural and functional skin changes are the most apparent feature of human aging, and the molecular mechanisms underlying skin aging resembles that seen in internal organs. Histologically aged skin is characterized by epidermal and subcutaneous atrophy, flattened dermal-epidermal junction, degeneration of collagen and elastic fibers, and reduction of skin vasculature [68]. Other hallmarks of skin aging include wrinkles, roughness, decreased elasticity, and hyperpigmentation.

At the cellular level, senescence is a process of gradual decrease in proliferative capacity, eventually leading to irreversible arrest at the G1 phase of the cell cycle and altered biosynthetic phenotype of the senescent cell (SASP-senescence-associated secretory phenotype) [69]. Three major mechanisms cooperating together are involved in the process of cellular aging: telomere shortening, insufficient DNA repair system, and accumulation of cellular damage in the form of byproducts generated in each and every cellular process. The term replicative senescence is related to shortening of the telomere length, which limits the number of cell divisions. On the other hand, stress-induced premature senescence occurs in response to DNA and other macromolecules damage caused by oxidative stress, oncogene activity, or chemical substances. Finally, the imperfectness of all biological processes leads to the accumulation of damage in postmitotic cells [70]. All the processes together subsequently contribute to genomic instability, triggering switch to SASP and cessation of cell division. Serially passaged cells show signs of cellular aging [71].

### 5.1. Sirtuin 1 Relevance in Intrinsic Aging

#### 5.1.1. Cutaneous Sirtuin 1 Expression Decreases with Age

First reports on sirtuin 1 relevance in skin aging dates back to 2006 [16,72]. SIRT1 is expressed in lower epidermis, upper dermis, and dermal–epidermal junction [72]. Sommer et al. [16], conducted a study on transgenic mice constitutively overexpressing ∆Np63 isoform, engaged in keratinocytes differentiation. The authors observed two main points: physical interaction between sirtuin 1 and ∆Np63, and correlation between its expression and senescent clinical phenotype of the rodents. The results suggest involvement of sirtuin 1 pathway in ∆Np63-induced aging model. Further studies on cell cultures and human tissues confirmed age-dependent downregulation of *SIRT1* gene expression [18,19,21]. Serial passage of human dermal fibroblasts, which is a model of replicative senescence, revealed decrease in extracellular matrix components accompanied by SIRT1 reduction [19]. Another study on SIRT1 expression in human skin biopsies, retrieved from females aged 20–67, revealed age-dependent decrease in SIRT1 in dermal fibroblasts [18]. In line with this, Golubstova et al. [21]. performed an analysis of tissue specimens from human fetuses and people aged between 20-week gestation and 85 years. The highest SIRT1 expression was found in fetal skin specimens. There were two sharp declines in SIRT1 expression across all studied population. First, in people under 20 years old compared to fetuses and second after the age of 40. Sirtuin 1 expression correlates with total fibroblast amount, and its proliferative activity and may be responsible for the development of skin aging symptoms. Aged fibroblasts lose metabolic and replicative activity leading to imbalanced turnover of extracellular matrix with decreased collagen, elastin, and hyaluronic acid content [73]. Aged skin is characterized by disrupted epidermal barrier function, partially due to decreased filaggrin expression, and filaggrin was shown to be expressed in SIRT1-dependent manner [22,74]. In the future, SIRT1 may be used as a marker of function of active fibroblasts and to verify effectiveness of active ingredients of antiaging cosmetic products.

#### 5.1.2. Premature Aging Related to Genetic Background

Extensive research on progeroid syndromes shed light on the molecular basis of aging processes. Progeroid syndromes are a group of clinically and genetically diversified disorders characterized by premature aging in more than one organ or tissue, and/or prominent susceptibility to harmful effects of ultraviolet radiation (UVR), resulting in increased incidence of skin cancers [75]. We can distinguish progeroid syndromes related to alterations in the nuclear architecture, and resulting from the mutations in DNA repair system.

Hutchinson-Gilford progeria syndrome (HGPS) is a rare fatal early-onset premature aging syndrome, affecting approximately 1 in 20 million children worldwide [76]. The underlying cause of the disease is a mutation-mediated alternative splicing of mRNAs issued from the *LMNA* gene, leading to the production of truncated lamin A precursors termed progerin [77]. Lamin A is a component of the nuclear matrix, and by regulation of the chromatin structure, is involved in replication, transcription, and DNA repair. The acetylation of histones and transcriptional factors acetylation is implicated in the regulation of gene expression. Given the fact that sirtuin 1 possesses histone deacetylase activity and progerin expression was found to raise during normal aging, gave researchers an impulse to investigate the interactions between those proteins [78]. Indeed, lamin A had been identified as a direct potent activator of SIRT1 [33]. In contrast, SIRT1 association with progerin was considerable reduced [33]. Interestingly, the study brought evidence that resveratrol, beyond direct SIRT1 activation, enhances lamin’s A stimulating effect. Moreover, mice harboring the *LMNA* mutations treated with resveratrol for four months had significantly increased survival, and displayed reversal of the progeroid phenotype [33]. Of note, in November 2020, the FDA approved the first drug for HGSP-lonafarnib-an inhibitor of farnesyltransferase which prevent the buildup of truncated protein [79].

Werner syndrome, also called adulthood-progeria, is caused by autosomal recessive mutations in *WRN* encoding helicase engaged in DNA repair [80]. Clinically it is characterized by the absence of the growth spurt seen early in puberty. In the further course of the disease several others hallmarks of aging develops, including accelerated atherosclerosis which is the leading cause of death in those patients. Small experimental study demonstrated involvement of sirtuin 1 in transcriptional regulation of *WRN* [26]. Sirtuin 1-dependent decline in WRN protein activity contributes to impaired DNA repair system during aging.

The efficiency of DNA repair system decreases with age [81], possibly due to reduced constitutive protein levels, contributing to chronical aging. Apart from that, mutations in the components of the repair system lead to multi-systemic premature aging with increased risk for cancer, and in some cases neurodegeneration [77,80,82]. It was proposed that neurogenerative phenotype is related specifically to mitochondrial dysfunction [28,32,33]. Noteworthy, the activation of SIRT1 by poly-ADP-ribose polymerase-1 (PARP1) inhibitors or NAD^+^ precursors were able to rescue mitochondrial defects, attenuate neurodegeneration, and extend lifespan in experimental models [28,32,33]. Moreover, a literature review revealed implication of sirtuin 1 in response to DNA damage by interaction with nucleotide excision repair (NER) system. NER system consists of two subpathways: global genomic (GG-NER) and transcription-coupled repair (TC-NER) [83]. The system is essential for repair damage related to chemical and physical agents, especially UV radiation. Direct absorption of UV light induces the formation of covalent bonds between DNA strands that distorts the helical structure [84]. The process of DNA repair occurs in sequential steps, beginning with the recognition of the site of helical distortion, followed by dual incision, repair synthesis, and ligation [83]. The first step is dependent on the XPA protein, which beyond recognition, is responsible for binding of the DNA excision repair 1 protein complex (ERCC1), and its recruitment to the DNA damage site [83]. The protein is regulated by posttranslational modifications by checkpoint kinase ATR or SIRT1 deacetylase under UV exposure [27,85]. SIRT1 augments NER pathway through XPA deacetylation, and promote interaction between XPA and ATR [27,29]. Transcription of XPC protein, a component of the preincision complex, was demonstrated to be regulated in SIRT1-dependent manner [30]. The disturbed interplay between sirtuin 1 and NER pathway is not only one of the mechanisms underlying cell vulnerability in genetic defects of the NER system, but probably also participates in normal aging and skin carcinogenesis.

#### 5.1.3. Progressive Tissue Fibrosis as a Hallmark of Aging

Several recent studies gave a new perspective on the pathophysiology of systemic sclerosis (SSc) which determines its perception as a premature aging syndrome. The main features of the disease involve fibrosis, vascular abnormalities, and dysregulation of the immune system [86]. TGFβ signaling plays a crucial role in the development of internal organ fibrosis, a hallmark of aging, and may be attenuated by SIRT1 activation in kidney and heart [87,88]. In line with this, recent findings confirmed the significance of sirtuin 1 in TGFβ-related fibrosis in SSc [23,24,25]. SIRT1 expression and protein level in skin samples obtained from patients with SSc and animal models were considerably decreased compared to healthy controls. As genetic and pharmacologic activation of SIRT1 substantially reduced skin fibrosis, it has the potential to introduce novel therapies in SSc and morphea.

### 5.2. Sirtuin 1 Relevance in Extrinsic Aging

#### 5.2.1. Sirtuin 1 Affects Ultraviolet Irradiation-Related Skin Changes by Numerous Targets

Chronological aging is eminently influenced by several environmental factors that according to Krutmann et al. [89] may be divided into following major categories: sun radiation, air pollution, tobacco smoke, stress, nutrition, lack of sleep, and temperature. It is estimated that up to 90% of symptoms of premature facial aging is related to cumulative exposure to sunlight [90]. A recent study investigating the epidermal expression of SIRT1 in samples obtained from healthy volunteers revealed significantly reduced expression in sun-exposed area compared to non-exposed skin [43]. The results suggest relevance of epigenetic regulation of gene expression in the pathogenesis of photoaging.

Photodamage is attributed to UVB (280–315 nm), UVA (315–400 nm), visible light (400–740 nm), and infrared radiation (740 nm–1 mm), with deeper skin penetration with the increasing wavelength [89]. UV radiation acts on cells through several mechanisms that interfere with each other and aggravate detrimental effects. The mechanisms involve release of proinflammatory cytokines, upregulation of collagenases, direct DNA oxidation, and indirect modifications of DNA, proteins, and lipids related to ROS production [91] (Figure 2).

Photoaging is commonly attributed to the UVA rays which penetrate to the dermis and are primarily responsible for alternations in human dermal fibroblasts and extracellular matrix. Although photoaging affects all three layers of the skin tissue, UV-related changes in the epidermis seems to be secondary to those in the dermal compartment [89].

Extracellular matrix (ECM) constitutes a highly dynamic tridimensional structure, filling intercellular spaces, built up of structural and functional proteins, glycoproteins and proteoglycans. It undergoes constant reconstruction, exhibiting significant alterations with the passage of time [92]. Decreased content of collagen, associated with aging, is a result of the imbalance between production and proteolytic degradation of the fibers [93]. Fragmentation and disorganization of the dermal ECM attenuates interaction between fibroblasts and collagen I [94]. The deprivation is the direct cause of the acquisition of senescent phenotype by fibroblast and may be in a vast majority reversed by enhancing structural support of the ECM. The theory is supported by the observation on skin biopsies retrieved from healthy individuals ≥70 years of age, treated with cross-linked hyaluronic acid injection [95]. The procedure stimulated fibroblast proliferation, expanded vasculature, and increased epidermal thickness. It highlights the pivotal role of the ECM turnover in the maintenance of skin tissue function.

Matrix metalloproteinases (MMPs) represent a group of 23 proteolytic enzymes that play a key role in physiologic and pathologic processes in human body [96]. They are essential for proper angiogenesis, apoptosis, wound healing, and scar formation. On the other hand, they facilitate tumor cell invasion and metastasis, or contribute to photoaging [73,96]. Of the 19 MMPs presented in human skin, only three were shown to be induced in response to UV radiation: MMP-1, MMP-3, and MMP-9 of which MMP-1 is responsible for initiation of collagen degradation [97]. The source of MMPs in the skin can be both epidermal keratinocytes and dermal fibroblasts [73]. The UV-related damage to the collagen and elastin ECM fibers is the cause of wrinkles as well as laxity and loss of skin firmness [73]. A study conducted by Ohuguchi et al. [35] demonstrated regulation of basal and stimulated expression of MMP-1 and MMP-3 by sirtuin 1 which proves its impact on intrinsic and extrinsic aging. Recent findings from experimental study on human keratinocytes and dermal fibroblasts revealed the ability of *Lactobacillus acidophilus* (KCCM12625P) to induce mRNA expression of *SIRT1* in parallel with decline in MMP-1 and elastase expression [37]. Similar results were obtained with the use of pyrroloquinoline quinone (PQQ), a potent antioxidant, known as longevity vitamin and activator of sirtuin 1 and 3 [39]. Specifically, UVA-irradiated HDFs preconditioned with PQQ exhibited reduced expression of β-galactosidase and MMPs (1 and 3).

Sirtuin 1 overexpression was found to enhance oxidative stress resistance via FOXO3α-promoted upregulation of the ROS superoxide dismutase 2 (SOD2) and catalase [40]. Experimental evidence showed considerably decreased sirtuin 1 in human dermal fibroblasts exposed to hydrogen peroxide [36,38]. Consistently with previous studies on other tissues, SIRT1 activators were able to inhibit apoptosis, and increase cell survival [36,38,98]. Taniguchi et al. [36] compared pretreatment with ascorbic acid and its stable derivative 2-*O*-α-glucopyranosyl-l-ascorbic acid (AA-2G) with regard to its protective effect against cellular damage and senescence [36]. Only AA-2G prevented the decrease in SIRT1 expression under cellular stress which rationalize the widespread use of stable vitamin C derivatives in the cosmetic industry. However, it must be noticed that in the described research the ascorbic acid also was able to retain fibroblasts proliferative capacity, but its effect was short-lasting and insufficient to obtain the desired results in clinical settings.

Recently, considerable effort has been put into exploring molecular mechanisms underlying the effect of SIRT1 on the UV-induced alterations in skin. Beyond increased oxidative stress resistance and enhanced NER repair system, sirtuin 1 reduces transcriptional activity of p53 and p16 proteins and its downstream pathways [40,41,42,44]. While generally cell survival is advantageous in terms of skin aging, chronic accumulation of damage and attenuation of p53-related cell cycle arrest may contribute to the survival of DNA damaged keratinocytes, and thus, skin carcinogenesis. UVB radiation is almost completely absorbed by the epidermis, and triggers generation of reactive oxygen species [99]. Epidermal SIRT1 expression is downregulated upon UVB exposure, and consequently the acetylated p53 protein activates apoptotic cell death [100]. Sublethal chronic exposure to UVB is insufficient to initiate apoptosis, but induces keratinocytes differentiation [101]. On the one hand, it promotes cancer cell survival, but on the other, is responsible for therapeutic effect of phototherapy on abnormal hyperproliferation of keratinocytes seen in psoriasis. Under natural conditions, the skin is exposed to UVR and high temperature at the same time, but those factors act oppositely on sirtuin 1 expression [100,102]. Several studies have shown that heat stress promotes skin carcinogenesis [102,103,104]. The issue of subsequent action of UVB and heat on skin cancer was raised in the study conducted by Calapre et al. [41]. The findings from the study are directly in line with previous findings on regulation of SIRT1 expression by UVR and high temperature [38,105]. Their experiment also found clear support for the SIRT1-mediated cell survival under heat, and concomitant heat and UVB exposure. In the presence of SIRT1 inhibitor the level of acetylated p53 protein, as well as apoptosis, were substantially elevated. To sum up, the posttranslational modification of p53 plays a role in UVB and heat induced cell survival and carcinogenesis. 

Based on human cancer-associated data, previously reported in the literature, SIRT1 serves as a tumor promoter [45]. Immunohistochemical analysis of sirtuin 1 expression in samples obtained from non-melanoma skin cancers and actinic keratosis have shown its overexpression in comparison to normal skin and benign tumors [45]. However, the *SIRT1* gene dosage also have a prominent impact on skin response to UVR [46]. The wild type and heterozygous for *Sirt1* mice are prone to skin cancer development due to reduced DNA repair and cell survival, whereas *Sirt1*-homodeficiency increases apoptosis signaling and decreases tumorigenesis, but sensitizes skin to solar injury [46]. Furthermore, SIRT1, through MMP2 regulation, is implicated in tumor metastasis [50,56]. Sirtuin 1 biological role in cancer is complex and depends on its targets in specific signaling pathways.

#### 5.2.2. Sirtuin 1 as a Promising ‘Weapon’ against Pollutant-Related Premature Aging

According to World Health Organization, 91% of the world’s population live in places where air quality exceeds guideline limits for major pollutants [106]. Advances in understanding air pollution on human health prompted global public health institutions to raise awareness of governments and individuals about environmental protection. The skin is directly exposed to air pollutants which may be smoothly absorbed to subcutaneous tissue, contributing to aggravation of several chronic inflammatory skin diseases and acceleration of skin aging [107]. Major air pollutants include polycyclic aromatic hydrocarbons, particulate matter, ozone, carbon monoxide, nitrogen, and sulfur dioxide [106]. All of them are associated with increased oxidative stress and extrinsic aging. Tobacco smoke remains major source of hazardous indoor air pollutant for both: active and secondhand smokers. Cigarette smoke is composed of over 7000 chemicals of which, at least, 70 are carcinogenic [108]. In addition to the obvious impact on the development of lung cancer and chronic obstructive pulmonary disease (COPD), the dramatic acceleration of natural skin aging by cigarette smoke is well documented [109]. The characteristic pattern of cigarette-related facial aging includes deep periorbital and perioral wrinkling, accompanied by orange skin discoloration, and occasionally large comedones [109]. The detrimental effect of smoking is driven by massive oxidative stress generation, antioxidant mechanisms impairment, and upregulation of matrix metalloproteinases [110,111]. So far, there are only indirect evidence for sirtuin 1 regulation of skin changes related to air pollutants. The assumptions about the role of SIRT1 might be driven from studies on COPD. Chronic inflammation, excessive oxidative stress, and increased activity of metalloproteinases underlie the pathogenesis of COPD. In details, SIRT1 overexpression may prevent airway inflammation, and oxidative stress induced by particulate matter [112,113,114]. Moreover, SIRT1 overexpression, by downregulation of MMP-1 and MMP-3, has beneficial effect on mice emphysema and human COPD [115]. Of note, it regulates exactly the same metalloproteinases that are critically involved in the imbalance of extracellular matrix turnover in the skin [116,117]. However, further research is needed to investigate the role of SIRT1 in skin aging related cigarette smoke and ambient pollutants.

### 5.3. Potential Interventions in Aging Process by Sirtuin 1 Activation

#### 5.3.1. Sirtuin 1 Mediates the Benefits of Caloric Restriction on Healthspan 

The most valuable non-pharmacologic intervention to improve lifespan is caloric restriction, and its beneficial effects are mediated by raised sirtuin 1 level [62]. The advantageous impact of CR is explained by the hormesis dose-response phenomenon [118,119]. The hypothesis assumes that low-intensity stressor stimulates cellular processes to enhance defense capacity, while high-intensity one, exceeding the threshold dose, leads to deleterious biological effects [120]. Doses within hormetic zone induce adaptive cellular responses which further improve tolerance to high-dose factors. The term should be equated with the pre- and postconditioning, previously reported in medical literature [120]. Cellular stress response, stimulated by fasting or its polyphenolic mimetics, upregulates the expression of vitagenes, a family of genes, consisting of heat shock proteins (HSPs), thioredoxin system, sirtuins, and superoxide dismutase (SOD). The equilibrium between long-term benefit from fasting compared with harm from insufficient caloric intake may vary across the population, and precise recommendations on energy intake restriction presumably cannot be ascertained.

Sirtuin 1 switches the energy source from glucose to lipids during fasting. Inhibition of glycolysis increases NAD^+^/NADH ratio [121], and sirtuin 1 activity in keratinocytes and dermal fibroblasts which translates into augmented differentiation of keratinocytes and prolonged life of the latter [17,22]. So far, direct assessment of CR on human skin has not been performed, however, long-term CR decreases glycation of extracellular proteins in rodents [122]. On the other hand, replicative senescence is not affected by fasting [123]. A study performed by De Cabo et al. [56] on cultured human fibroblast, treated with serum obtained from rodents on various dietary regimens, investigated alterations in several aging biomarkers. The researchers found that serum from rats fed on CR significantly delays the downregulation of SIRT1 and increase in β-galactosidase associated with senescence phenotype. However, the impact of fasting on human skin aging has not been evaluated so far.

#### 5.3.2. Sirtuin 1 Activators for the Topical Treatment of Premature Aging Symptoms

From the standpoint of applicability of sirtuin 1 anti-aging properties in dermatology and cosmetology, the most profitable would be to use topical activators of the protein. In 2007, a study on biopeptides derived from yeast Kluyveromyces revealed increase in SIRT1 expression in human dermal fibroblasts and keratinocytes with concomitant decrease in β-galactosidase [48]. The results were confirmed by the objective improvement in skin aging symptoms after four-week application of a topical formula enriched in the yeast biopeptides [48]. However, among studied so far activators, resveratrol has drawn the most attention. It is a phytoalexin from the stilbene family of polyphenols synthetized in the skin of grapes, blueberries, raspberries, mulberries, and peanuts in response to infection or physical injury [124]. Whereas the phytoalexin concentration in plants is noxious for microorganisms, it is harmless to humans, and may induce adaptive stress response in compliance with hormesis concept. A large amount of data demonstrated health benefits of resveratrol, including the maintenance of cutaneous functions and delay of senescence symptoms [125,126,127,128,129]. Furthermore, it protects skin from various tissue damaging stimuli: UV radiation [130] or excessive oxidative stress, generated by tobacco smoke [131]. The biological effects of resveratrol mentioned above were also reported in vivo with topical application. Numerous lines of evidence proved protective effect against UVR-induced damage [132,133,134] and increased antioxidant capacity [135,136]. The major limitation of topical resveratrol applicability is fast isomerization and poor water solubility [137,138] what prompted the researchers to investigate a novel stable and potent resveratrol derivative [139,140]. In regards of skin aging, the data on the results of oral application are conflicting. A placebo-controlled trial of a dietary supplement containing polyphenols demonstrated visual improvement in multiple age-related skin changes as well as decreased systemic oxidative stress [141]. In contrast, an antioxidant cocktail, composed of trans-resveratrol, selenium, and vitamin E and C, increased antioxidant capacity without obvious skin improvement in a study on 60 healthy volunteers [142]. The discrepancies may be partially explained by multicomponent composition of the supplements in both studies. The same interpretation problem may arise from the study on topical formulation containing resveratrol, retinol, niacinamide, and hexylresorcinol [143]. Although the study confirmed effectiveness in combating numerous skin aging symptoms, it cannot attribute this effect directly to resveratrol. Similarly, due to involvement of multiple mechanisms, not all effects of the resveratrol are associated with SIRT1 activation. In reference to sirtuin 1 pathway, resveratrol regulates keratinocytes differentiation and proliferation, promotes wound healing, decreases keloid formation, participates in antioxidant defense, and UVR protection [144].

Garlic (*Allium sativum*) is the source of many valuable bioactive compounds and has been used as natural remedy since ancient times. Traditionally, it is considered as natural antibiotic due to the content of allicin that is responsible for the majority of its antimicrobial activity [145]. Moreover, garlic extract is also a potent antioxidant what made it applicable in dermatology. Topical application of garlic extract is beneficial for the treatment of inflammatory skin diseases, keloids, wound healing, as well as skin aging and photoprotection [146]. The study on human keratinocytes exposed to UVB confirmed that some of these properties are mediated by SIRT1 pathway [57]. Garlic pretreatment reduced oxidative stress, inflammation, and aging biomarkers [57]. Several other studies investigated the effect of pretreatment with multiple biological compounds on UVR-related alterations in human dermal fibroblasts [49,52,58]. The examined botanical compounds, including resveratrol and *Epilobium angustifolium* extract, were able to restore UV-induced downregulation of SIRT1 and decrease the activity of metalloproteinases [49,52]. 

Several other Chinese herbal medicine-derived compounds were found to attenuate oxidative injury by the upregulation of the SIRT1 pathway [59,60]. Kumar et al. [59] pointed out the potential antiaging application of *Antrodia cinamomea*, an endemic medicinal mushroom from Taiwan, possessing hepatoprotective effects. The fungus contains several bioactive compounds of which antcin M have been reported to reduce hyperglycemia-induced oxidative stress in dermal fibroblasts. *Tremella fuciformis*, another example of edible mushroom, has numerous beneficial features including: anti-inflammatory, antioxidant, antitumor, and antiaging [60]. It has been used for a long time as an important part of Chinese cuisine, dietary supplement, and antiaging ingredient of topical creams, serums, and facial preparations; however, the involvement of SIRT1 mechanism was described recently [60].

In contrast, some bioactive compounds, isolated from plants and fungi, delayed cellular senescence of HDFs without affecting the SIRT1 expression and activity [51,54]. For instance, epigallocatechin gallate (EGCG), a polyphenol extracted from green tea, significantly suppressed the p53 acetylation but had no effect on SIRT1 [51]. It is worthy to mention that experimental concentration of the EGCG, providing antiaging effects, greatly exceeded normal dietary consumption. Similar results were obtained by Takata et al. [54] in a study aimed to determine the effect of saikokeishito (TJ-10) on HDFs senescence. Saikokeishito is a traditional Japanese herbal medicine (Kampo) consisting of nine herbal components: *Bupleurum* root, *Pinellia tuber*, *Scutellaria* root, Jujube fruit, Ginger rhizome, ginseng root, cinnamon bark, peony root, and *Glycyrrhiza* root [54]. TJ-10 has been widely prescribed against infectious diseases, chronic pancreatitis (due to suppression of the TGFβ expression), and other gastrointestinal diseases as well as an antiepileptic drug (activation of the GABA-A receptors). The results presented by Takata et al. suggests that it also participates in dermal senescence; however, it is unclear at the time if oral intake of the TJ-10 may bring about objective improvement in skin aging.

The conflicting results on oxidative stress reduction by food-derived compounds may be due to polyphenols instability in incubation medium that enhances the production of hydrogen peroxide and affect SIRT1 activity [147]. Because the dietary polyphenols undergo rapid metabolism, form glucuronide and sulfate conjugates in the liver, the bioactivity of botanical compounds in the human body may differ from the results obtained in the in vitro studies [147]. However, from the standpoint of hormesis response, it is not essential to reach experimental concentrations, exceeding dietary intake, and sometimes high doses of phytochemicals may be toxic [148]. Growing body of evidence indicates that hormetic mechanisms explain similar health benefits achieved by numerous phytochemicals intake. The key element of cellular stress response is activation of the nuclear factor-erythroid 2 p45-related factor 2 (Nrf2) which binds to antioxidant response element (ARE), a promoter of several genes crucial for redox homeostasis. ARE encodes cytoprotective proteins with diversified functions: antioxidant, anti-inflammatory, repair or removal of damaged macromolecules, conjugation, and glutathione homeostasis [120]. Menendez et al. [53] demonstrated that anticancer and antiaging activity of the complex of polyphenols naturally presented in extra virgin olive oil is related to Nrf2 signaling. Hydroxytyrosol, one of the main polyphenols of the olive oil, has been recently shown to prevent neurodegeneration in nematodes and rodents [149,150]. The authors of the studies postulated that Nrf2 signaling pathway and hormesis response may be responsible for the observed effects. Considering common pathophysiologic background of neurodegenerative disorders, age-related diseases, and skin aging nutraceuticals may be effective and safe therapeutic targets to improve healthy aging.

## 6. Conclusions

The past few decades have witnessed rapidly growing interest in the topic of skin aging. Research advances in the field of molecular mechanisms involved in both intrinsic and extrinsic aging may translate into novel effective therapeutic antiaging interventions. SIRT1 regulates the proliferation and differentiation of the major cell types of the skin. It mediates antiaging effects through the target molecules, participating in the maintenance of genome stability, oxidative stress response, extracellular matrix turnover, and regulation of the cell cycle. The same mechanisms are involved in the skin carcinogenesis, however, SIRT1 may serve as tumor suppressor or oncogene, depending on the gene dosage and downstream pathways. Experimental studies support the conclusion that SIRT1 may prevent UVR-related premature aging and skin cancer development. Modulation of SIRT1 activity may constitute a novel approach to photoprotection, and hold a great potential for improving outcomes in patients with skin cancers. In addition, small interventional studies in humans demonstrated encouraging results of topical SIRT1 activators in reducing premature aging symptoms. Although SIRT1 has been extensively researched, further studies are needed to explore the topic of SIRT1 activity in terms of skin changes, induced by environmental pollutants, which help to exploit the full potential of this longevity protein.

## Figures and Tables

**Figure 1 cells-10-00813-f001:**
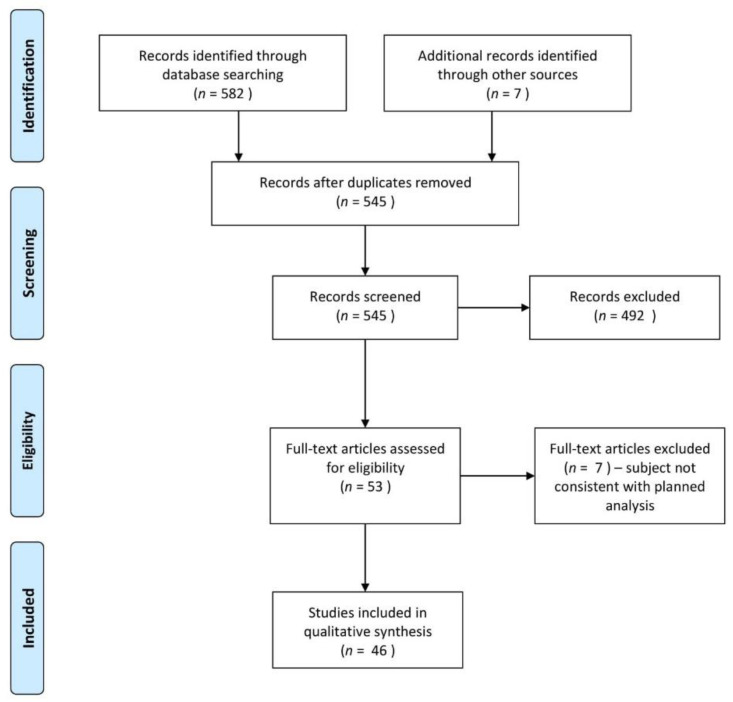
Search strategy and results.

**Figure 2 cells-10-00813-f002:**
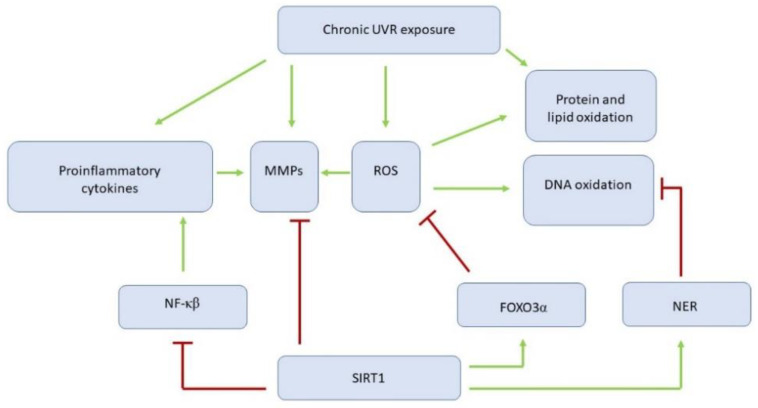
Multidirectional impact of ultraviolet radiation on skin aging. (UVR, ultraviolet radiation; MMPs, matrix metalloproteinases; ROS, reactive oxygen species; NF-ĸβ, nuclear factor kappa-light-chain-enhancer of activated B cells; FOXO3α, Forkhead Box O3 alpha; NER, nucleotide excision repair system; SIRT1, sirtuin 1).

**Table 1 cells-10-00813-t001:** Summary of the studies on SIRT1 relevance in intrinsic aging.

Author	Year	Population	Key Observation
**Intrinsic Aging–Basic Information**			
Sommer et al. [16]	2006	∆Np63α transgenic mice Normal lung epithelial cells transfected with a vector containing∆Np63α	transgenic mice exhibited an accelerated aging phenotype in the skin accompanied by a decrease in longevity correlated with levels of SIRT1. In cell culture beta-galactosidase accumulation and typical senescent morphology was rescued by SIRT1.
Yang et al. [17]	2011	HDFs, Hs68 cell culture exposed to 2-DG and DHEA	2-DG, but not DHEA, at non-cytotoxic concentrations extends lifespan in parallel with increased intracellular NAD^+^ levels and SIRT1 activities
Kalfalah et al. [18]	2014	Skin biopsies of females aged 20–67	Age-dependent decrease in SIRT1
Kim et al. [19]	2015	Passaged HDFs culture	SIRT 1 is down regulated by increasing passage of HDFS.
Lee et al. [20]	2016	HaCaT keratinocytes	Melatonin-induced autophagy play a protective role through SIRT1 pathway against skin cell damage as a result of hydrogen peroxide-induced cell death.
Golubtsova et al. [21]	2017	Skin biopsies retrieved from deceased donors: fetuses at pregnant age 20–40 week, people from birth to 85 years old	Age-related decrease in sirtuin 1 content in HDFs is correlated with age-dependent decrease in fibroblasts proliferation. The highest level of SIRT1 is found between 20- to 40-week of pregnancy.
Sutter et al. [22]	2019	NHEKs and N/TERT-1	Decreased glucose metabolism increases keratinocytes differentiation by SIRT1 activation.
**Systemic Sclerosis**			
Wei et al. [23]	2015	Skin biopsy samples of healthy adults and patients with SSc. *Sirt1*^−/−^ and wild-type mouse embryonic fibroblasts	Reduced SIRT1 expression and protein level in SSc skin biopsy samples compared to healthy. Activation of *Sirt1* attenuated fibrosis, while inhibition had profibrotic effects.
Zerr et al. [24]	2016	Skin biopsies of patients with SSc and healthy volunteers *Sirt1*^−/−^ and wild-type mice	SIRT1 is decreased in TGF- β-dependent manner in patients with SSc and in experimental fibrosis. SIRT1 activation enhances the profibrotic effects of TGF-β with increased Smad reporter activity, elevated transcription of TGF-β target genes and raised release of collagen. *Sirt1* KO inhibited TGF-β/SMAD signaling and reduced release of collagen in fibroblasts. *Sirt1*^−/−^ mice were less susceptible to fibrosis
Zhu et al. [25]	2017	Skin biopsy specimens of SSc patients and healthy controls Mice treated with BLM	*SIRT1*, activated by RSV, ameliorated cutaneous inflammation and fibrosis in BLM- induced scleroderma. The enhancement of mTOR expression in the skin of the mice was significantly inhibited by Sirt1 activation.

Abbreviations: SIRT1—sirtuin 1; HDFs—human dermal fibroblasts; 2-DG—2-deoxyglucose; DHEA—dehydroepiandrosterone; NHEKs—Neonatal normal human epithelial keratinocytes; N-TERT1—immortalized keratinocytes; SSc—systemic sclerosis; RSV—resveratrol; BLM—bleomycin; Mtor—mammalian target of rapamycin kinase.

**Table 2 cells-10-00813-t002:** Summary of the studies on SIRT1 relevance in progeroid syndromes.

Author	Year	Mutation	Population	Key Observations
Kayho et al. [26]	2008	WRN	*Sirt1*-heterozygous mice, homozygous mutant offspring *Sirt1*^−/−^ HEK293T cells	CR of rats led to a simultaneous increase in the level of WRN and SIRT1 protein. WRN was decreased in *Sirt1*-deficient mice and HEK293T cells, treated with sirtuin inhibitors.
Fan et al. [27]	2010	XPA	HeLa, H1299, HEK293T cells and XPA-deficient fibroblasts	UVR augments the XPA and SIRT1 interaction, which leads to cell survival. Phosphorylation works in acute response to UV-induced damage, while acetylation is involved in DNA repair complex Downregulation of SIRT1 delays the removal of CPD, but no 6-4PPs lesion. SIRT1 functions as a tumor suppressor.
Fang et al. [28]	2014	XPA	In silico on-line database www.mitodb.com, accessed on 6 April 2021 (Scheibye-Knudsen et al., 2013)	In XPA, CS, and AT, SIRT1 attenuation leads to decreased mitophagy through the depression of PGC-1α and UCP2. The mitochondrial abnormalities appear to be caused by decreased activation of the NAD^+^-SIRT1-PGC-1α axis triggered by hyperactivation of the DNA damage sensor PARP1.
Jarrett et al. [29]	2018	XPA	A375 melanoma cells UV-irradiated	SIRT1-dependent deacetylation of XPA augments cAMP-enhanced NER.
Ming et al. [30]	2010	XPC	Human skin tumor samples UV irradiated	Inhibition of SIRT1 impairs global genome NER through suppressing the transcription of XPC in a SIRT1 dependent manner. SIRT1 levels are significantly reduced inhuman skin tumors from Caucasian patients. SIRT1 acts as a tumor suppressor.
Velez-Cruz et al. [31]	2013	XPD subunit of TFIIH	Human primary fibroblast	Transcriptional arrest upon UVR in XP-D/CS cells results from gene repression mediated by SIRT1 and may be restored with a Sirt1-specific inhibitor or downregulation by siRNA.
Scheibye-Knudsen et al. [32]	2014	CSB	Four-month-old mice SV40-transformed CS1AN cells Bristol N2 (WT) and csb-1 worms	Premature aging results from aberrant PARP activation due to deficient DNA repair leading to decreased SIRT1 activity and mitochondrial dysfunction. B-hydroxybutyrate levels are increased by the high-fat diet, and b-hydroxybutyrate, PARP inhibition, or NAD^+^ supplementation can activate SIRT1 and rescue CS-associated phenotypes.
Fang et al. [33]	2016	ATM	Primary neurons, nematodes, *Atm*^−/−^ mice	A-T laboratory animal models exhibit NAD^+^ depletion and impaired SIRT1 activity. NAD^+^ replenishment improves lifespan and healthspan in worms and mice and ameliorates A-T phenotypes through upregulation of mitophagy and DNA repair.
Liu et al. [34]	2012	LMNA	HEK293 cells, mouse embryonic fibroblasts, HDFs derived from HGPS patients and healthy individuals, cells harboring *LMNA* mutations, *Zmpste24*/mice, bone marrow stromal cell and hematopoietic stem	Lamin A activates SIRT1 deacetylase. Resveratrol enhances SIRT1 activity by increasing its interaction with lamin A. Prelamin A or progerin has significantly reduced association with SIRT1 in cells. SIRT1 deacetylase activity is compromised in progeroid cells. Resveratrol alleviates progeroid features and extends life span in progeria mice.

Abbreviations: WRN—Werner syndrome ATP-dependent helicase; XPA—xeroderma pigmentosum A gene/protein; XPC—xeroderma pigmentosum C gene/protein; XPD—xeroderma pigmentosum D gene/protein; TFIIH—transcription factor IIH; CSB—Cockayne syndrome B gene/protein; ATM—ataxia telangiectasia mutated kinase serine/threonine; LMNA—lamin A/C gene; CR- caloric restriction; UVR—ultraviolet radiation; CPD—cyclobutane pyrimidine dimers; 6-4PPs—pyrimidine 6-4 pyrimidone photoproducts; PGC1α—Peroxisome Proliferator-Activated Receptor Gamma Coactivator 1 Alpha; PARP1—Poly[ADP-ribose] polymerase 1; UCP2—mitochondrial uncoupling protein 2; NER—nucleotide excision repair complex; A-T—ataxia telangiectasia; SIRT1—sirtuin 1; HDFs—human dermal fibroblasts; HGPS—Hutchinson Gilford progeria syndrome.

**Table 3 cells-10-00813-t003:** Summary of the studies on SIRT1 relevance in extrinsic aging.

Author	Year	Population	Key Observation
Extrinsic Aging			
Ohguchi et al. [35]	2010	HDFs treated with SIRT1 inhibitor, activator and IL-1β	SIRT1 negatively regulates transcription of *MMP-1* and *MMP-3* and controls both basal and IL-1β-induced *MMP* expression
Taniguchi et al. [36]	2012	HDFs exposed to H_2_O_2_, AA, AA-2G	H_2_O_2_ reduced SIRT1 in HDFs. Pretreatment with AA-2G significantly inhibits reduction, whereas AA had no effect.
Lim et al. [37]	2020	HaCaT, HDFs and B16F10 cells exposed to UVB irradiation	AL has an antiwrinkle activity in damaged skin and can inhibit melanogenesis, regulates baseline *MMP* expression and induces collagen production in HDFs. AL inhibits elastase and MMP-1 and induces type 1 procollagen. AL increased the expression of SIRT1
**Extrinsic Aging–UV Irradiation**			
Cao et al. [38]	2009	HaCaT, p53 wild-type mouse, MEFs and p53 knockout MEFs	UVR and H2O2 downregulate SIRT1. RSV protects against cell death, whereas SIRT inhibitors enhance it
Zhang et al. [39]	2015	HDFs exposed to UVA irradiation	PQQ reduces the expression of senescence markers MMP1, MMP3 and beta-galactosidase by up-regulation of SIRT1
Chung et al. [40]	2015	HS27 culture cell and HR1 hairless mouse exposed to UVB irradiation	SIRT1 overexpression protects fibroblasts from UVB-induced cell cycle arrest by p53 deacetylation. SIRT1 thorough FOXO3α increases resistance to the oxidative stress.
Calapre et al. [41]	2017	ex vivo skin models, taken from non-sun exposed skin of healthy donors and NHEKs exposed to UVB plus heat	SIRT1 mediates UVB plus heat induced survival of DNA damaged keratinocytes by: decrease in p-53 acetylation and downregulation of its downstream pathways, including: BAX, ERCC1, XPC Increase in Ki67.
Lei et al. [42]	2018	Primary HDFs obtained from foreskins of healthy human donors aged 5–20 years exposed to UVA irradiation	Fluorofenidon alleviates HDFs senescence by inhibiting the mTOR and increasing SIRT1
Ding et al. [43]	2018	Epidermis isolated from skin biopsies obtained from the outer forearm and the buttock of healthy females	Lower expression of HDAC1 and SIRT1 in sun-exposed skin compared with matched non-exposed skin
Li et al. [44]	2020	HaCaT exposed to UVA irradiation	α-l-Hexaguluroic acid hexasodium salt (G6) increases mitochondrial metabolism, alleviates oxidative stress, reverses the downregulation of SIRT1 and pGC-1a expression levels
**UV-Related Carcinogenesis**			
Hida et al. [45]	2007	Immunohistochemical staining for SIRT1 expression in 87 cases of skin tumors and 20 normal skin samples.	Sun-exposed and sun-protected skin regions did not differ in SIRT1 expression. SIRT1 was overexpressed in all samples of AK, BD, SCC and BCC. SIRT1 overexpression may have some relevance to the early stage of skin carcinogenesis.
Ming et al. [46]	2015	*Sirt1* cKO and cHet, WT mice, and NHEKs exposed to UVB irradiation. Human skin samples of SCC.	*Sirt1* cHET promotes UVB-induced skin tumorigenesis, whereas cKO *Sirt1* suppresses skin tumor development but sensitizes the mice to chronic solar injury. In mouse skin, *Sirt1* is haploinsufficient for UVB-induced DNA damage repair. SIRT1 is downregulated in parallel with XPC in human SCC. cKO deletion of *Sirt1* augments p53 acetylation and sensitizes the epidermis to UVB-induced apoptosis in vivo, while heterozygous has no such effect. UVB induced tumor formation in *Sirt1* WT and *Sirt1* cHet mice but not in *Sirt1* cKO mice.
Brandl et al. [47]	2019	human BCC human and murine normal skin	The c-MYC-NAMPT-DBC1-SIRT1 positive feedback loop may play a role in the development of BCCs.

Abbreviations: SIRT1—sirtuin1; HDFs—human dermal fibroblasts; IL-1β—interleukin 1β; MMP1- matrix metalloproteinase 1, MMP3—matrix metalloproteinase 3; AA—ascorbic acid; AA-2G—2-O-α-glucopyranosyl-l-ascorbic acid; UVR—ultraviolet radiation; UVB—ultraviolet B; UVA—ultraviolet A; AL—Lactobacillus acidophilus KCCM12625P; MEFs—mouse embryonic fibroblasts; RSV—resveratrol; H2O2—hydrogen peroxide; PQQ—pyrroloquinoline quinone; NHEKs—neonatal normal human epithelial keratinocytes; BAX—BCL2 Associated X, Apoptosis Regulator; ERCC1—Excision Repair 1, Endonuclease Non-Catalytic Subunit; XPC—xeroderma pigmentosum C protein; FOXO3α—Forkhead Box O3; mTOR—mammalian target of rapamycin kinase; HDAC1—histone deacetylase 1; pGC-1a—peroxisome proliferator-activated receptor gamma coactivator 1-alpha; AK—actinic keratosis; BD—Bowen’s disease; SCC—squamous skin carcinoma; BCC—basal cell carcinoma; cKO—homozygous knockout; cHet—heterozygous deletion; WT—wild type; DBC1—Deleted in Breast Cancer 1; NAMPT—Nicotinamide phosphoribosyltransferase.

**Table 4 cells-10-00813-t004:** Summary of the studies on SIRT1 activators with reference to skin outcomes.

Dietary and Supplementary Interventions				
Author	Year	Population	Substance	Key Observations
Moreau et al. [48]	2007	HDFs skin samples of females aged 37 to 64.	emulsion enriched with 1% of the yeast Kluyveromyces biopeptides	SIRT1 is expressed in epidermis and dermis. SIRT1 expression is greater in proliferating compared to differentiated keratinocytes. Kluyveromyces biopeptides stimulate SIRT1 expression in keratinocytes and HDFs, decrease beta-galactosidase and enhance DNA integrity. Objective improvement in multiple skin aging symptoms after topical use of the formulation.
Lee et al. [49]	2010	HDFs and keratinocytes isolated from human skin sample. Mouse skin model.	Resveratrol Metformin	*SIRT1* activation by RSV and metformin inhibits MMP-9 expression under UVR exposure
Park et al. [50]	2012	HDFs	Spermidine	*SIRT1* gene expression was increased by spermidine Spermidine inhibits activity and expression of MMP-2
Han et al. [51]	2012	HDFs	EGCG	EGCG at high concentration prevents serial passage and H2O2-induced senescence in HDFs via suppression of the p53 without affecting the SIRT1 activity
Ruszova et al. [52]	2013	HDFs Healthy volunteers aged 40–50	Epilobium angustifolium (EA) extract	EA extract downregulated MMP-1,-3 and TIMP-1,-2 by repeatedly UV irradiated HDFs. EA extract diminished SIRT1 downregulation dampened by UV-irradiation and decreased UV-induced erythema formation in vivo.
Menendez et al. [53]	2013	HDFs	EVOO	EVOO prevents age-related changes in the cell size, morphological heterogeneity, arrayed cell arrangement and senescence-associated β-galactosidase staining of HDFs by the activation of ER stress and the unfolded protein response, spermidine and polyamine metabolism, SIRT1 and NRF2 signaling.
Takata et al. [54]	2014	HDFs	Saikokeishito	Saikokeishito protects HDFs from premature senescence by hydrogen peroxide, but had no effect on SIRT1 expression.
Watanabe et al. [55]	2015	*Sod1*^−/−^ mice primary dermal fibroblasts from skin tissue of *Sod1*^−/−^ neonates	MSE	Orally MSE and RSV treatment reversed the skin thinning associated with increased oxidative damage in the *Sod1*^−/−^ mice. MSE and RSV normalized gene expression of *Col1a1* and p53 and upregulated gene expression of *Sirt1* in skin tissues.
De Cabo et al. [56]	2015	HDFs	Serum from rats fed on caloric restriction (40%) versus ad libitum diets	CR serum delays the passage-induced senescent phenotype, reduces SA-β-Gal and MMP-2 activity. CR serum prevents SIRT1 downregulation. Overexpression of SIRT1 in late passage human fibroblasts resulted in delayed senescent growth arrest. KO of *SIRT1* in early passage cells enhanced MMP-2 activation.
Kim [57]	2016	UVB-exposed human keratinocytes, HaCaT cells	Garlic	Garlic pretreatment attenuated in a dose-dependent manner the production of ROS, proinflammatory cytokines and MMP-1 protein expressions. SA-β-gal and SIRT1 activity were ameliorated by garlic treatment.
Wahedi et al. [58]	2016	UVB-irradiated HaCaT and HDFs	Juglone	Juglone restored the expression of SIRT1 and Pin1 in almost dose dependent manner in irradiated cells. Juglone treatment upregulated SIRT1 in unirradiated skin cells.
Kumar et al. [59]	2016	HDFs	Antcins (*Antrodia cinnamomea*)	Antcin M protects HDFs from hyperglycemia-induced cell-cycle arrest and oxidative injury by upregulation of SIRT1 expression.
Shen et al. [60]	2017	HDFs	Tremella fuciformis polysaccharide	TFPS relieves H2O2-induced HDFs injury by attenuating oxidative stress and cell apoptosis by upregulation of the SIRT1 pathway.
Wang et al. [61]	2019	HaCaT	Angelica polysaccharide	AP alleviates LPS-induced injury through upregulating SIRT1 expression and then activating Nrf2/ HO-1 pathway but inactivating NF-κB pathway

Abbreviations: SIRT1—sirtuin 1; HDFs—human dermal fibroblasts; RSV—resveratrol; MMP-9—matrix metalloproteinase 9; MMP-2—matrix metalloproteinase 2; MMP-1—matrix metalloproteinase 1; UVR—ultraviolet radiation; UVB—ultraviolet B; ECCG—epigallocatechin3-O-gallate; H2O2—hydrogen peroxide; TIMP—tissue inhibitor of metalloproteinases; EVOO—complex of polyphenols naturally present in extra virgin olive oil; ER—endoplasmic reticulum; NRF2—nuclear factor erythroid 2-related factor 2; MSE—Melinjo (Gnetum gnemon Linn) seed extract; Pin1—peptidyl-prolyl cis-trans isomerase NIMA-interacting 1; CR—caloric restriction; AL—ad libitum; KO—knockout; TFPS—Tremella fuciformis polysaccharide; AP—Angelica polysaccharide; LPS—lipopolysaccharides; NF-κB—nuclear factor kappa-light-chain-enhancer of activated B cells; ROS—reactive oxygen species.

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
