# Peer review of "Sirtuin 1 and Skin: Implications in Intrinsic and Extrinsic Aging—A Systematic Review"

_cells, 2021, doi:10.3390/cells10040813_

Round 1
Reviewer 1 Report
The authors reviewed the literature regarding the sirtuin 1 as a target to treat or prevent skin ageing.
The section 5.3.2 on sirtuin 1 activators for the topical treatment would be more interesting and helpful for the reader if putative sirtuin 1 activators were described in more details.
In a general manner, English should be improved, in particular a better use of punctuation.
Minor points :
- give some description on saikokeishito.
- page 8 lines 156-157 : a third mechanism is involved in cellular ageing : the theory of cellular damage accumulation, which is a generalisation of the first theory of aging described by Denham Harman. It would be important to mention it. Finally all three mechanisms join together and combine to explain ageing.
- page 10, lines 272-273 : (1) typing mistake : "740 nm - 1 µm"; (2) the end of the sentence ("with deeper ... detrimental effects") is not understandable; please clarify.
- page 11, line 279 (Figure 2 legend) : typing mistake : "NF-kB".
page 14, line 459 : typing mistake : "differ from"
Reviewer 2 Report
In numerous experimental models, natural antioxidants such as sulforaphane (SFN) or Hydroxytyrosol (HD) is shown to induce hormetic dose responses that are not only common but display endpoints of biomedical and clinical relevance. These hormetic responses are mediated via the activation of nuclear factor erythroid- derived 2 (Nrf2) antioxidant response elements (AREs) and, as such, are characteristically biphasic, well integrated, concentration/dose dependent, and specific with regard to the targeted cell type and the temporal profile of response. In experimental disease models, the polyphenol-induced hormetic activation of Nrf2 was shown to effectively reduce the occurrence and severity of a wide range of human-related pathologies, including Parkinson's disease, Alzheimer's disease, stroke, age-related ocular damage, chemically induced brain damage, and renal nephropathy, amongst others, while also enhancing stem cell proliferation. Interestingly, the mechanistic profile of SFN or HD is similar to that of numerous other hormetic agents, indicating that activation of the Nrf2/ARE pathway is probably a central, integrative, and underlying mechanism of hormesis itself. The Nrf2/ARE pathway provides an explanation for how large numbers of agents that both display hormetic dose responses and activate Nrf2 can function to limit age-related damage, the progression of numerous disease processes, and chemical- and radiation- induced toxicities. These findings extend the generality of the hormetic dose response to include SFN and many other chemical activators of Nrf2 that are cited in the biomedical literature and therefore have potentially important public health and clinical implication. Thus, Interplay and coordination of redox interactions with endogenous and exogenous antioxidant defence systems is an emerging area of reserach interest in anticancer and antidegenerative therapeutics. Moreover, particular attention has been given to providing an assessment of the quantitative features of the dose-response relationships and underlying mechanisms that could account for the biphasic nature of the hormetic response after exposure to redox active agents, such as free radical oxygen species and their impact in inflammatory/antinflammatory pathways. The hormetic dose response should be seen as a reliable feature of the dose response for oxygen free radicals and their redox regulated transcriptional factors as well as antioxidant compounds and appears to have an important impact on brain pathophysiology and stress resistance mechanisms to oxidative and inflammatory insult and neurodegenerative damage. This is an interesting paper. The study is well-conceived and well-executed. This reviewer is satisfied with the significance of this study, the care in which the study was performed, and the implications of the results for human health. However, although the results presented are convincing, the work raises some concerns which will need to be addressed. The questions posed are of extremely high interest, but the paper does not give adequate definitive information, therefore pending addressing some major question is possible to accept for publication. Minor concerns: 1. Preconditioning signal leading to cellular protection through Hormesis is an important redox dependent aging-associated to free radicals species accumulation, inflammatory responses involved in neurodegenerative/ neuroprotective mechanisms. This aspect should be highlighted in the discussion and references properly added (See Calabrese et al., 2010, Antiox. Redox Signal 13,1763; Siracusa et al., Antioxidants 2020, 9(9):E824). 2. Given the relationship between polyphenol compounds, redox status and the vitagene network and its possible biological relevance in neuroprotection, Authors while interpetrating results should discuss appropriately this aspect and make proper connection with emerging principles of hormesis ( Brunetti et al., Int J Mol Sci. 2020 Apr 8;21(7):2588).
